# Linear epitope mapping in the E and NS1 proteins of dengue and Zika viruses: Prospection of peptides for vaccines and diagnostics

Victor Hugo Aquino[1,2☯]*, Marcilio J. Fumagalli[3], Angélica Silva[3], Bento Vidal de Moura Negrini[4], Alejandra Rojas[5], Yvalena Guillen[5], Cynthia Bernal[5], Luiz Tadeu Moraes Figueiredo[3☯]*

1 Immunology Department, Research Institute for Health Sciences, National University of Asuncion, San Lorenzo, Central, Paraguay, 2 Department of Clinical Analyses, Toxicology and Food Sciences, School of Pharmaceutical Sciences, University of Sao Paulo, Ribeirao Preto, Sao Paulo, Brazil, 3 Virology Research Center, Ribeirao Preto Medical School, University of Sao Paulo, Ribeirao Preto, Sao Paulo, Brazil, 4 School of Medicine, Federal University of Sao Carlos, Sao Carlos, Sao Paulo, Brazil, 5 Production Department, Research Institute for Health Sciences, National University of Asuncion, San Lorenzo, Central, Paraguay

☯ These authors contributed equally to this work.
* vhaquino@iics.una.py (VHA); ltmfigue@fmrp.usp.br (LTMF)

**Data Availability Statement:** The datasets generated for peptide microarray analyses have been deposited in the Gene Expression Omnibus

## Abstract

The arrival of the Zika virus (ZIKV) in dengue virus (DENV)-endemic areas has posed challenges for both differential diagnosis and vaccine development. Peptides have shown promise in addressing these issues. The aim of this study was to identify the linear epitope profile recognized by serum samples from dengue and Zika patients in the E and NS1 proteins of DENV and ZIKV. This cross-sectional study included individuals of all ages with laboratory-confirmed DENV and ZIKV infections, who were selected through convenience sampling. The serum samples from dengue and Zika patients detected epitopes evenly distributed across the viral proteins in a peptide microarray platform. However, several epitopes were located within "epitope hotspots", characterized by clusters of peptides recognized in more than 30% of the sub-arrays analyzed using individual or pooled serum samples. The serum samples from dengue and Zika patients showed a high level of cross-reactivity with peptides in the DENV and ZIKV proteins. Analysis using an additional peptide microarray platform, which contained peptides selected based on the results of the initial screening, revealed that two DENV and one ZIKV peptide, highly specific to their related viruses, were located within the epitope hotspots; however, they presented low detection rates (32.5, 35.0, and 28.6%, respectively). In addition, two DENV peptides detected at similarly high rates by both dengue and Zika patients were also found within the epitope hotspots. These hotspots contain several immunodominant epitopes that are recognized by a larger number of individuals when compared to 15-amino acid (aa) sequence peptides. Thus, epitope hotspots may have greater potential to serve as antigens in diagnostic tests and vaccine development than peptides composed of only 15 amino acids.

(GEO) (GSE235045, https://www.ncbi.nlm.nih.gov/geo/query/acc.cgi?acc=GSE235045).

**Funding:** This study was financially supported by Fundação de Amparo à Pesquisa do Estado de São Paulo, FAPESP, Brazil (grant numbers 2017/09194-3 and 2019/26119-0), and Consejo Nacional de Ciencia y Tecnología, CONACYT, Paraguay (grant number PIRT19-1). The funders had no role in study design, data collection and analysis, decision to publish, or preparation of the manuscript.

**Competing interests:** The authors have declared that no competing interests exist.

## Introduction

Dengue and Zika are two mosquito-borne diseases of global concern that mainly affect tropical and subtropical regions [1–3]. The transmission of dengue virus (DENV) and Zika virus (ZIKV) occurs primarily through the bites of female *Aedes aegypti* mosquitoes. Infections with any of the four DENV serotypes (DENV-1, -2, -3, and -4) can be asymptomatic or cause illness, with clinical presentations ranging from mild to severe forms of the disease. The two primary clinical forms are dengue with or without warning signs and severe dengue [4]. Generally, the initial clinical manifestation is a sudden onset of fever, which may be associated with headache, myalgia, arthralgia, retro-ocular pain, and exanthema. Some patients may progress to severe forms of the disease, presenting increased vascular permeability, resulting in the loss of vascular plasma fluid that can lead to hypovolemic shock, which, in some cases, may be fatal. Until recently, human cases of ZIKV infection occurred sporadically; however, this virus began to gain notoriety after its introduction in Brazil in 2015, specifically in the Northeast [5–7], from where it subsequently disseminated throughout the entire country [8]. This virus was responsible for large epidemics in Brazil associated with increased microcephaly cases in newborns, Guillain-Barré syndrome, and other neurological complications [9–11]. Dengue is endemic in the Americas, with sustained epidemics dating back to the 1970s [12]. As a result, many people living in the Americas have been immunized against DENV. The introduction of ZIKV in DENV-endemic areas posed challenges for the differential diagnosis of the two viral infections and the development of candidate vaccines. Both viruses belong to the *Flavivirus* genus, within the *Flaviviridae* family; therefore, they share several immunological antigens that can lead to cross-immune responses [13–15]. Most of the countries in South America include the Yellow fever virus (YFV) vaccine in their vaccination regimens. YFV also belongs to the *Flavivirus* genus, like the dengue and Zika viruses. This similarity can pose additional challenges in serological tests, as cross-reactivity may occur during the diagnosis of dengue and Zika.

Peptides have shown significant potential in advancing the development of highly specific diagnostic tests and vaccines [16–18]. In the present study, we unveiled the epitope profile recognized by dengue and Zika patients in the E and NS1 proteins of both DENV and ZIKV, highlighting certain peptides that show promise for the development of serological tests for the differential diagnosis of dengue and Zika infections, as well as for the production of candidate vaccines.

## Materials and methods

### Study design

This cross-sectional study was conducted from March 2017 to December 2022. The eligible population, selected through convenience sampling, consisted of individuals of all ages with suspected dengue or Zika disease. Inclusion criteria required participants to have laboratory confirmation of DENV or ZIKV infection. Laboratory tests included reverse transcription-polymerase chain reaction (RT-PCR), IgG/IgM tests, and/or the 50% plaque reduction neutralization test (PRNT50). Exclusion criteria encompassed participants with negative IgG tests. The eligible control population comprised healthy individuals vaccinated against yellow fever, who had negative serological tests for dengue and Zika. Medical records served as the primary source of information regarding the presence of the disease, relying on laboratory test results. This assessment method applied to both the virus-infected groups and the healthy control group.

The Virology Research Center of the Ribeirao Preto Medical School, University of Sao Paulo, and the Virology Laboratory of the School of Pharmaceutical Sciences, University of Sao Paulo, served as the centers for participant recruitment. All participants provided signed written consent for their inclusion in the study. In cases where participants were children under 18

years of age, parent/guardian consent was obtained, and the children themselves also signed the written consent form. In addition, archived serum samples from de-identified collections, dating from April 2011 to February 2020, located at the Virology Research Center of the Ribeirao Preto Medical School, University of Sao Paulo, the Virology Laboratory of the School of Pharmaceutical Sciences, University of Sao Paulo, and the Production Department of the Research Institute for Health Sciences, National University of Asuncion, Paraguay, were also included in the study. For the archived samples, the ethics committees waived the need for consent.

The participants were selected through convenience sampling. Due to budget limitations, the sample size was determined by the number of available microarray slides.

### Ethics statement

This study was reviewed and approved by the Ethics Committees of the School of Pharmaceutical Sciences of Ribeirao Preto (SPSRP), University of Sao Paulo (USP), Brazil (CEP/FCFRP n ˚ 1845548), and the Research Institute for Health Sciences of the National University of Asuncion, Paraguay (P08/2018).

### Peptide microarray for epitope mapping

Representative isolates of DENV-1 (GenBank AKQ00011), DENV-2 (GenBank AGX15379), DENV-3 (GenBank AFK83760), DENV-4 (GenBank AEW50183), and ZIKV (GenBank AMA12085) were selected based on previous studies [19–23] to design the peptide microarray platform. The array contained linear peptides with overlapping sequences covering the entire viral E (393 aa without the transmembrane region) and NS1 proteins. The proteins were linked and elongated with neutral GSGSGSG linkers at the C- and N-termini and translated into overlapping peptides. The peptides (n = 1898) were 15 aa long, with a peptide-peptide overlap of 13 aa. The peptide microarray slide contained two identical sub-arrays (PEPperPRINT, Heidelberg, Germany). Each sub-array included the viral peptides printed in duplicated spots and two control peptides in 126 spots each: YPYDVPDYAG and KEVPALTAVETGAT from the influenza virus and poliovirus, respectively.

### Peptide immunodetection

The peptide microarray slide was incubated for 15 min at room temperature in standard buffer (PBS, pH 7.4, 0.05% Tween 20), then in blocking buffer (PBS, pH 7.4, 0.05% Tween 20, 1% BSA) with shaking (140 rpm) for 60 min at room temperature, and finally in staining buffer (standard buffer plus 10% blocking buffer) with shaking (140 rpm) for 15 min at room for temperature for equilibration. The sub-arrays were incubated with individual or a pool of up to four (equal volumes) serum samples (diluted at 1:80 in staining buffer, pH 7.4) overnight at 4˚C. Afterward, the arrays were washed 3x1 min at 140 rpm with the standard buffer, followed by incubation with the secondary antibody (Goat anti-Human IgG Fc Cross-Adsorbed Secondary Antibody—DyLight 650, Invitrogen, USA, diluted at 1:5000 in staining buffer) for 30 min with shaking (140 rpm) at room temperature, in the dark. The arrays were subsequently washed 3x1 min with the standard buffer with shaking (140 rpm). Finally, the slide was dipped two times into the dipping buffer (1 mM Tris, pH 7.4) and dried carefully in an air stream.

### Data collection and analysis

The array images were obtained with the Axon GenePix 4000B scanner (Molecular Devices, USA) or InnoScan 710 (Innopsys, France), using a 635-nm laser and 10-μm resolution. Median fluorescence intensity for each spot, with local background subtraction, was computed using the GenePix Pro 7 (Molecular Devices, USA) or Mapix (Innopsys, France) software.

Fluorescence intensity values <1 were converted to 0. The fluorescence intensity values were then normalized against the negative controls (HA peptides). This normalization involved dividing the fluorescence intensity value of each spot by the mean fluorescence intensity of the negative control spots. Normalized fluorescence intensity values <1 were converted to 1 and log2-transformed to reduce variability. The final fluorescence intensity value for each peptide consisted of the mean of the fluorescence intensity values of the duplicated peptide spots.

## Mapping of linear epitopes in the E and NS1 proteins of DENV and ZIKV

The linear epitopes were mapped using serum samples from dengue (n = 19) and Zika (n = 28) patients in 16 and 11 arrays, respectively. In addition, serum samples from healthy individuals (n = 4), who had negative serological tests for dengue and Zika but were vaccinated against yellow fever, were analyzed in 3 arrays and served as controls. The fluorescence intensity threshold for each peptide was determined as the mean fluorescence intensity value plus two times the standard deviation of the mean intensity obtained with the control serum samples [24]. Peptides with fluorescence intensities exceeding their threshold values were considered as specifically detected by the serum samples from the dengue and Zika patients. The peptides detected by the sera of dengue and Zika patients enabled the identification of epitopes. An epitope refers to the amino acid sequence shared by adjacent overlapping peptides detected within a serum sample. This microarray included 15-aa-long peptides with a peptide-peptide overlap of 13 aa. Therefore, when two overlapping adjacent peptides were detected, the epitope was 13 aa long; when three overlapping adjacent peptides were detected, the epitope was 11 aa long, and so on. Thus, this platform can identify epitopes measuring 13, 11, 9, 7, 5, and 3 aa long. If a serum sample detected a single peptide, we considered the entire 15-aa peptide as an epitope.

## Peptide microarray for specificity analysis

After analyzing the results from the initial microarray platform, we selected several 15-aa peptides to design an additional peptide microarray slide. This microarray slide consisted of five identical sub-arrays, each containing duplicated spots of viral peptides, the influenza virus YPYDVPDYAG peptide in 38 spots, and the poliovirus KEVPALTAVETGAT peptide in 36 spots (PEPperPRINT, Heidelberg, Germany). The influenza virus and poliovirus peptides served as negative and positive controls in the immunoassays, respectively. Each sub-array was evaluated using individual serum samples obtained from dengue (n = 40) and Zika (n = 21) patients, as well as uninfected controls (n = 3). In order to compare the peptide detection rates obtained with the serum samples from dengue and Zika patients, we constructed a contingency (2 × 2) table and analyzed the data using Fisher's exact test [25]. All values were two-tailed, and the statistical significance threshold was set at $p < 0.05$.

## Blast

The BLASTP 2.14.0+ program [26] was used to analyze the similarities between virus-specific peptides and those demonstrating high detection rates with other virus strains. The peptide sequences were compared with strains of each virus species using the default algorithm parameters, considering a maximum of 1000 target sequences.

# Results

## Study population

This study included serum samples from 48 dengue patients, 28 Zika patients, and five healthy individuals. The control group consisted of healthy individuals with negative serological tests

for dengue and Zika and who had been vaccinated against yellow fever. The mean age was 33 and 34.4 years for the dengue and Zika patients, respectively, and 24 years for the healthy individuals. Among the dengue patients, 70.8% were female, while 75% were male in the Zika patient group. All healthy individuals were female. The demographic characteristics of the participants and the laboratory data used for diagnosis are shown in Table 1.

**Table 1. Demographic and laboratory characteristics of the study participants.**

| Characteristics | Dengue patients (n = 48) | Zika patients (n = 28) | Healthy individuals (n = 5) |
|---|---|---|---|
| Mean age ± SD (years) | 33 ± 18.1 | 34.4 ± 7.6 | 24 ± 10.9 |
| **Sex** | | | |
| Male | 14 (29.2%) | 21 (75%) | 0 |
| Female | 34 (70.8%) | 7 (25%) | 5 (100%) |
| **DENV-RT-PCR** | | | |
| Positive | 48 (100%) | - | |
| Negative | 0 | - | |
| Not determined | - | 28 (100%) | |
| DENV Type | | | |
| DENV-1 | 2 (4.2) | - | |
| DENV-2 | 8 (16.7) | - | |
| DENV-3 | 6 (12.5%) | - | |
| DENV-4 | 16 (33.3) | - | |
| DENV | 16 (33.3) | - | |
| **ZIKV-RT-PCR** | | | |
| Positive | - | 2 (7.1%) | |
| Negative | - | 0 | |
| Not determined | 48 (100%) | 26 (92.9%) | |
| **DENV-IgG** | | | |
| Positive | 28(58.3%) | 0 | 0 |
| Negative | 2 (4.2%) | 20 (71.4%) | 5 (100%) |
| Not determined | 18 (37.5%) | 8 (28.5%) | - |
| **DENV-IgM** | | | |
| Positive | 18 (37.5%) | 0 | 0 |
| Negative | 12 (25%) | 20 (71.4%) | 5 (100%) |
| Not determined | 18 (37.5%) | 8 (28.5%) | - |
| **ZIKV-IgG** | | | |
| Positive | - | 20 (71.4%) | 0 |
| Negative | - | 0 | 5 (100%) |
| Not determined | 48 (100%) | 8 (28.5%) | - |
| **ZIKV-IgM** | | | |
| Positive | - | - | 0 |
| Negative | - | - | 5 (100%) |
| Not determined | 48 (100%) | 28 (100%) | - |
| **DENV-PRNT50** | | | |
| Titer: <40 | - | 28 (100%) | - |
| Titer: >160 | - | 0 | - |
| Not determined | 48 (100%) | - | - |
| **ZIKV-PRNT50** | | | |
| Titer: <40 | - | 0 | - |
| Titer: >160 | - | 28 (100%) | - |
| Not determined | 48 (100%) | - | - |

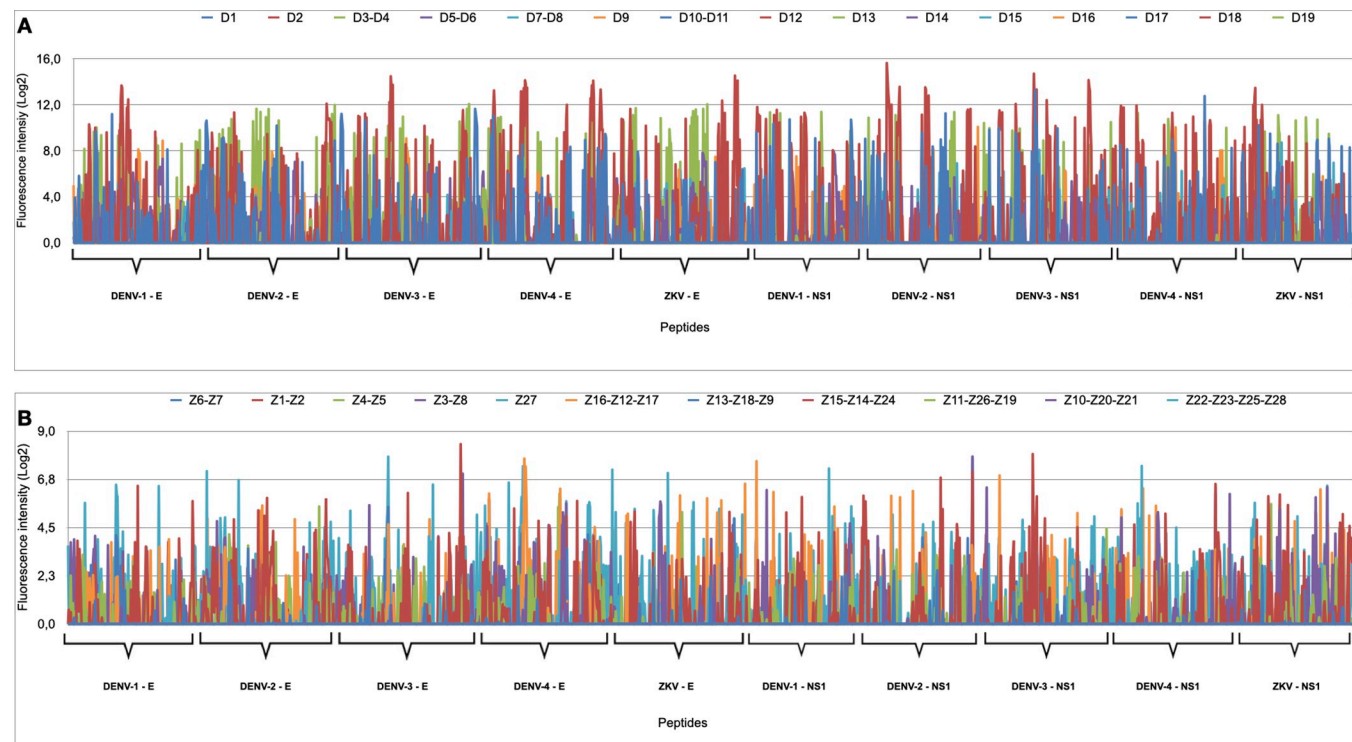

**Fig 1.** Peptide profile detected by the serum samples from dengue (A) and Zika (B) patients in the E and NS1 proteins of DENV-1, DENV-2, DENV-3, DENV-4, and ZIKV. The y-axis quantifies the intensity of detected peptides (depicted on the x-axis) by the patient samples, represented by vertical bars.

## Epitope mapping

A peptide microarray platform, specifically designed for epitope mapping in the E and NS1 proteins of DENV-1, DENV-2, DENV-3, DENV-4, and ZIKV, was subjected to analysis using individual or pooled serum samples obtained from dengue (19 serum samples in 15 sub-arrays) and Zika (28 serum samples in 11 sub-arrays) patients. According to Fig 1, the serum samples detected peptides distributed evenly throughout the viral proteins, indicating no specific preference for particular regions. Moreover, there was a high degree of cross-reactivity, as the serum samples exhibited similar detection rates for peptides derived from unrelated viruses when compared to those from their corresponding related viruses.

The peptides detected by the serum samples, with the exception of those containing part of the linker sequence (GSGSGSG), provided the epitope identification (Tables 2, S1 and S2). In each array, the serum samples detected a range of 60 to 303 epitopes, with sizes ranging from 3 to 15 aa, with the latter being more frequently observed. The amino acid sequences of the detected epitopes exhibited a high degree of variability (S1 and S2 Tables).

The detection rate for each peptide in the arrays varied from 0 to 86.7% and 0 to 90.9% for the serum samples from dengue and Zika patients, respectively. When analyzing the peptide detection rates, we identified epitope hotspots constituted by clusters of peptides that consistently exceeded a 30% detection rate (Fig 2 and S3 Table).

## Peptide specificity analysis

The peptides' specificity was further analyzed using an additional peptide microarray containing several peptides (n = 248) selected from the initial screening (S4 Table). These peptides

included those found within the hotspots, outside the hotspots, and undetected peptides. Each sub-array analyzed with individual serum samples from dengue (n = 40) and Zika (n = 21) patients provided information on the specificity of the serum samples. When comparing the peptide detection rates, three DENV peptides were significantly more frequently detected by patients with dengue than by those with Zika, and one ZIKV peptide was detected significantly more frequently by patients with Zika than by those with dengue (Tables 3 and S5).

The peptides TQGEPSLNEEQDKRF, TQTVGPWHLGKLEID, and LELDPPFGDSYIVIG were located within epitope hotspots (Table 4), whereas the peptide VHRQWFLDLPLPWLP was not.

In contrast, our findings revealed comparable detection rates between serum samples from dengue and Zika patients for most of the DENV and ZIKV peptides examined. Among these, we identified five peptides with high detection rates (exceeding a minimum arbitrary threshold of 40%) by both dengue and Zika patients (Table 5), two of which (WEVEDYGFGVFTTNI and LELDFDLCEGTTVVV) were located within the epitope hotspots (Table 6).

**Table 2. The number of peptides and epitopes detected by the serum samples from dengue and Zika patients in the E and NS1 proteins of DENV-1, DENV-2, DENV-3, DENV-4, and ZIKV.**

| Serum samples | Patient code | | Protein E | | | | | Protein NS1 | | | | | |
|---|---|---|---|---|---|---|---|---|---|---|---|---|---|
| | | | DENV-1 (n = 197) | DENV-2 (n = 197) | DENV-3 (n = 197) | DENV-4 (n = 197) | ZIKV (n = 201) | DENV-1 (n = 176) | DENV-2 (n = 176) | DENV-3 (n = 176) | DENV-4 (n = 176) | ZIKV (n = 176) | TOTAL |
| Dengue patients | D1 | P* | 32,0 | 43 | 27 | 45 | 22 | 17 | 17 | 20 | 24 | 23 | 270 |
| | | E# | 22,0 | 23 | 16 | 24 | 18 | 14 | 14 | 14 | 20 | 19 | 184 |
| | D2 | P* | 48,0 | 47 | 46 | 61 | 43 | 44 | 41 | 47 | 38 | 40 | 455 |
| | | E# | 20,0 | 24 | 26 | 24 | 25 | 21 | 16 | 19 | 21 | 23 | 219 |
| | D3-D4 | P* | 28,0 | 36 | 37 | 41 | 31 | 14 | 21 | 12 | 17 | 27 | 264 |
| | | E# | 11,0 | 13 | 14 | 16 | 20 | 5 | 13 | 8 | 11 | 15 | 126 |
| | D5-D6 | P* | 64,0 | 72 | 60 | 72 | 62 | 62 | 56 | 50 | 52 | 63 | 613 |
| | | E# | 25,0 | 29 | 31 | 27 | 26 | 31 | 21 | 25 | 23 | 25 | 263 |
| | D7-D8 | P* | 29,0 | 24 | 28 | 34 | 32 | 24 | 26 | 16 | 21 | 26 | 260 |
| | | E# | 13,0 | 11 | 15 | 15 | 17 | 13 | 13 | 13 | 16 | 18 | 144 |
| | D9 | P* | 21,0 | 27 | 20 | 19 | 14 | 18 | 15 | 13 | 19 | 7 | 173 |
| | | E# | 11,0 | 14 | 8 | 9 | 9 | 13 | 13 | 10 | 12 | 6 | 105 |
| | D10-D11 | P* | 46,0 | 38 | 42 | 37 | 28 | 11 | 20 | 20 | 19 | 19 | 280 |
| | | E# | 19 | 18 | 26 | 17 | 20 | 9 | 15 | 16 | 14 | 12 | 166 |
| | D12 | P* | 38 | 32 | 17 | 34 | 25 | 25 | 35 | 26 | 28 | 23 | 283 |
| | | E# | 23 | 17 | 13 | 24 | 24 | 16 | 21 | 16 | 21 | 20 | 195 |
| | D13 | P* | 27 | 39 | 30 | 44 | 33 | 19 | 16 | 16 | 13 | 21 | 258 |
| | | E# | 10 | 16 | 14 | 22 | 17 | 11 | 16 | 11 | 13 | 11 | 141 |
| | D14 | P* | 41 | 59 | 50 | 58 | 52 | 36 | 46 | 45 | 33 | 42 | 462 |
| | | E# | 22 | 19 | 19 | 25 | 17 | 19 | 23 | 20 | 19 | 22 | 205 |
| | D15 | P* | 30 | 50 | 45 | 44 | 54 | 35 | 43 | 36 | 41 | 46 | 424 |
| | | E# | 17 | 24 | 26 | 21 | 37 | 25 | 22 | 21 | 22 | 31 | 246 |
| | D16 | P* | 23 | 38 | 16 | 23 | 22 | 15 | 27 | 22 | 21 | 28 | 235 |
| | | E# | 15 | 20 | 14 | 13 | 16 | 15 | 15 | 10 | 15 | 16 | 149 |
| | D17 | P* | 10 | 30 | 14 | 15 | 18 | 15 | 19 | 17 | 15 | 14 | 167 |
| | | E# | 9 | 12 | 8 | 10 | 10 | 14 | 14 | 11 | 13 | 9 | 110 |
| | D18 | P* | 39 | 54 | 34 | 51 | 40 | 23 | 26 | 24 | 20 | 26 | 337 |
| | | E# | 22 | 23 | 21 | 20 | 24 | 13 | 14 | 13 | 17 | 13 | 180 |
| | D-19 | P* | 27 | 29 | 30 | 30 | 33 | 19 | 17 | 17 | 16 | 19 | 237 |
| | | E# | 19 | 27 | 22 | 17 | 21 | 13 | 13 | 13 | 13 | 13 | 171 |

(*Continued*)

**Table 2.** (Continued)

| Serum samples | Patient code | | Protein E | | | | | Protein NS1 | | | | | TOTAL |
|---|---|---|---|---|---|---|---|---|---|---|---|---|---|
| | | | DENV-1 (n = 197) | DENV-2 (n = 197) | DENV-3 (n = 197) | DENV-4 (n = 197) | ZIKV (n = 201) | DENV-1 (n = 176) | DENV-2 (n = 176) | DENV-3 (n = 176) | DENV-4 (n = 176) | ZIKV (n = 176) | |
| Zika patients | Z6-Z7 | P* | 5 | 12 | 9 | 5 | 9 | 7 | 9 | 8 | 4 | 7 | 75 |
| | | E# | 5 | 8 | 7 | 4 | 7 | 5 | 7 | 7 | 3 | 7 | 60 |
| | Z1-Z2 | P* | 43 | 57 | 43 | 49 | 37 | 38 | 35 | 45 | 40 | 61 | 448 |
| | | E# | 22 | 25 | 21 | 21 | 20 | 21 | 18 | 20 | 19 | 23 | 210 |
| | Z4-Z5 | P* | 36 | 55 | 36 | 41 | 34 | 35 | 31 | 37 | 39 | 39 | 383 |
| | | E# | 18 | 21 | 20 | 20 | 17 | 23 | 18 | 19 | 14 | 17 | 187 |
| | Z3-Z8 | P* | 24 | 34 | 30 | 41 | 30 | 26 | 31 | 23 | 26 | 41 | 306 |
| | | E# | 16 | 18 | 22 | 25 | 16 | 14 | 17 | 16 | 15 | 17 | 176 |
| | Z27 | P* | 26 | 40 | 40 | 41 | 45 | 29 | 21 | 25 | 29 | 25 | 321 |
| | | E# | 14 | 19 | 19 | 26 | 30 | 23 | 13 | 20 | 19 | 21 | 204 |
| | Z16-Z12-Z17 | P* | 32 | 40 | 30 | 50 | 34 | 32 | 32 | 34 | 29 | 32 | 345 |
| | | E# | 22 | 21 | 18 | 26 | 25 | 23 | 20 | 24 | 22 | 22 | 223 |
| | Z13-Z18-Z9 | P* | 22 | 27 | 23 | 31 | 21 | 24 | 18 | 20 | 18 | 20 | 224 |
| | | E# | 15 | 15 | 11 | 14 | 14 | 17 | 8 | 14 | 6 | 14 | 128 |
| | Z15-Z14-Z24 | P* | 26 | 36 | 27 | 29 | 20 | 27 | 18 | 27 | 13 | 20 | 243 |
| | | E# | 14 | 20 | 15 | 15 | 17 | 15 | 10 | 16 | 10 | 12 | 144 |
| | Z11-Z26-Z19 | P* | 16 | 31 | 22 | 37 | 19 | 16 | 16 | 15 | 17 | 17 | 206 |
| | | E# | 13 | 17 | 13 | 17 | 14 | 12 | 14 | 10 | 14 | 11 | 135 |
| | Z10-Z20-Z21 | P* | 48 | 46 | 29 | 48 | 52 | 33 | 26 | 35 | 36 | 41 | 394 |
| | | E# | 24 | 25 | 16 | 19 | 30 | 17 | 15 | 20 | 20 | 20 | 206 |
| | Z22-Z23-Z25-Z28 | P* | 72 | 79 | 51 | 81 | 59 | 54 | 49 | 64 | 55 | 53 | 617 |
| | | E# | 30 | 33 | 28 | 36 | 33 | 31 | 27 | 30 | 29 | 26 | 303 |

*Peptides
#Epitopes.

## Peptide similarity analysis

The BLAST analysis showed that the DENV-specific peptides from Table 3 exhibited high similarity to all four DENV serotypes, while displaying comparatively lower similarity to the ZIKV strains (Table 7). Similarly, the ZIKV-specific peptide demonstrated substantial sequence identity with several ZIKV strains, while exhibiting minor similarity with DENV serotypes, except for DENV-4. The high-detection-rate peptides (Table 4) were found exclusively in DENV, and their high similarity to other strains from all four DENV serotypes supports this observation. These peptides also displayed substantial similarity to ZIKV strains, except for LELDFDL-CEGTTVVV and PIVTDKEKPVNIETE, which showed median identity rates of up to 50.60% with them. However, it is noteworthy that these peptides contain conserved epitopes shared between DENV and ZIKV, such as the LEIRFE, GTKVHV, PVITE, and LELD epitopes from ZIKV, a fact that may explain the high detection rates observed in Zika patients for both peptides.

## Discussion

Understanding the epitope profiles recognized by the humoral immune response for specific antigens, such as DENV and ZIKV proteins, is vital for developing candidate vaccines and diagnostic tests. Epitopes can be either linear or conformational [27]. Linear epitopes consist

| Virus-Protein | Sequence | Detection rate (%) | Virus-Protein | Peptides | Detection rate (%) | Virus-Protein | Sequence | Detection rate (%) | Virus-Protein | Sequence | Detection rate (%) |
|---|---|---|---|---|---|---|---|---|---|---|---|
| DENV-1-E | GSGSGSGMRCVGIGN | 20,0 | DENV-2-E | GSGSGSGMRCIGISN | 6,7 | DENV-3-E | GSGSGSGMRCVGVGN | 6,7 | DENV-4-E | GSGSGSGMRCVGVGN | 6,7 |
| DENV-1-E | GSGSGMRCVGIGNRD | 0,0 | DENV-2-E | GSGSGMRCIGISNRD | 6,7 | DENV-3-E | GSGSGMRCVGVGNRD | 0,0 | DENV-4-E | GSGSGMRCVGVGNRD | 0,0 |
| DENV-1-E | GSGMRCVGIGNRDFV | 6,7 | DENV-2-E | GSGMRCIGISNRDFV | 26,7 | DENV-3-E | GSGMRCVGVGNRDFV | 6,7 | DENV-4-E | GSGMRCVGVGNRDFV | 6,7 |
| DENV-1-E | GMRCVGIGNRDFVEG | 20,0 | DENV-2-E | GMRCIGISNRDFVEG | 40,0 | DENV-3-E | GMRCVGVGNRDFVEG | 20,0 | DENV-4-E | GMRCVGVGNRDFVEG | 0,0 |
| DENV-1-E | RCVGIGNRDFVEGLS | 20,0 | DENV-2-E | RCIGISNRDFVEGVS | 60,0 | DENV-3-E | RCVGVGNRDFVEGLS | 6,7 | DENV-4-E | RCVGVGNRDFVEGVS | 0,0 |
| DENV-1-E | VGIGNRDFVEGLSGA | 6,7 | DENV-2-E | IGISNRDFVEGVSGG | 46,7 | DENV-3-E | VGVGNRDFVEGLSGA | 6,7 | DENV-4-E | VGVGNRDFVEGVSGG | 20,0 |
| DENV-1-E | IGNRDFVEGLSGATW | 26,7 | DENV-2-E | ISNRDFVEGVSGGSW | 53,3 | DENV-3-E | VGNRDFVEGLSGATW | 33,3 | DENV-4-E | VGNRDFVEGVSGGAW | 26,7 |
| DENV-1-E | NRDFVEGLSGATWVD | 46,7 | DENV-2-E | NRDFVEGVSGGSWVD | 66,7 | DENV-3-E | NRDFVEGLSGATWVD | 46,7 | DENV-4-E | NRDFVEGVSGGAWVD | 40,0 |
| DENV-1-E | DFVEGLSGATWVDVV | 66,7 | DENV-2-E | DFVEGVSGGSWVDIV | 60,0 | DENV-3-E | DFVEGLSGATWVDVV | 20,0 | DENV-4-E | DFVEGVSGGAWVDLV | 46,7 |
| DENV-1-E | VEGLSGATWVDVVLE | 46,7 | DENV-2-E | VEGVSGGSWVDIVLE | 53,3 | DENV-3-E | VEGLSGATWVDVVLE | 13,3 | DENV-4-E | VEGVSGGAWVDLVLE | 40,0 |
| DENV-1-E | GLSGATWVDVVLEHG | 33,3 | DENV-2-E | GVSGGSWVDIVLEHG | 40,0 | DENV-3-E | GLSGATWVDVVLEHG | 46,7 | DENV-4-E | GVSGGAWVDLVLEHG | 33,3 |
| DENV-1-E | SGATWVDVVLEHGSC | 60,0 | DENV-2-E | SGGSWVDIVLEHGSC | 46,7 | DENV-3-E | SGATWVDVVLEHGGC | 53,3 | DENV-4-E | SGGAWVDLVLEHGGC | 40,0 |
| DENV-1-E | ATWVDVVLEHGSCVT | 26,7 | DENV-2-E | GSWVDIVLEHGSCVT | 40,0 | DENV-3-E | ATWVDVVLEHGGCVT | 26,7 | DENV-4-E | GAWVDLVLEHGGCVT | 13,3 |
| DENV-1-E | WVDVVLEHGSCVTTM | 53,3 | DENV-2-E | WVDIVLEHGSCVTTM | 53,3 | DENV-3-E | WVDVVLEHGGCVTTM | 33,3 | DENV-4-E | WVDLVLEHGGCVTTM | 26,7 |
| DENV-1-E | DVVLEHGSCVTTMAK | 20,0 | DENV-2-E | DIVLEHGSCVTTMAK | 6,7 | DENV-3-E | DVVLEHGGCVTTMAK | 13,3 | DENV-4-E | DLVLEHGGCVTTMAQ | 13,3 |
| DENV-1-E | VLEHGSCVTTMAKNK | 0,0 | DENV-2-E | VLEHGSCVTTMAKNK | 0,0 | DENV-3-E | VLEHGGCVTTMAKNK | 0,0 | DENV-4-E | VLEHGGCVTTMAQGK | 0,0 |
| DENV-1-E | EHGSCVTTMAKNKPT | 6,7 | DENV-2-E | EHGSCVTTMAKNKPT | 6,7 | DENV-3-E | EHGGCVTTMAKNKPT | 13,3 | DENV-4-E | EHGGCVTTMAQGKPT | 6,7 |
| DENV-1-E | GSCVTTMAKNKPTLD | 6,7 | DENV-2-E | GSCVTTMAKNKPTLD | 6,7 | DENV-3-E | GGCVTTMAKNKPTLD | 6,7 | DENV-4-E | GGCVTTMAQGKPTLD | 6,7 |
| DENV-1-E | CVTTMAKNKPTLDIE | 20,0 | DENV-2-E | CVTTMAKNKPTLDFE | 20,0 | DENV-3-E | CVTTMAKNKPTLDIE | 26,7 | DENV-4-E | CVTTMAQGKPTLDFE | 20,0 |
| DENV-1-E | TTMAKNKPTLDIELL | 20,0 | DENV-2-E | TTMAKNKPTLDFELI | 46,7 | DENV-3-E | TTMAKNKPTLDIELQ | 46,7 | DENV-4-E | TTMAQGKPTLDFELT | 46,7 |
| DENV-1-E | MAKNKPTLDIELLKT | 40,0 | DENV-2-E | MAKNKPTLDFELIKT | 46,7 | DENV-3-E | MAKNKPTLDIELQKT | 26,7 | DENV-4-E | MAQGKPTLDFELTKT | 60,0 |
| DENV-1-E | KNKPTLDIELLKTEV | 20,0 | DENV-2-E | KNKPTLDFELIKTEA | 53,3 | DENV-3-E | KNKPTLDIELQKTEA | 26,7 | DENV-4-E | QGKPTLDFELTKTTA | 53,3 |
| DENV-1-E | KPTLDIELLKTEVTN | 40,0 | DENV-2-E | KPTLDFELIKTEAKQ | 53,3 | DENV-3-E | KPTLDIELQKTEATQ | 33,3 | DENV-4-E | KPTLDFELTKTTAKE | 46,7 |
| DENV-1-E | TLDIELLKTEVTNPA | 20,0 | DENV-2-E | TLDFELIKTEAKQPA | 53,3 | DENV-3-E | TLDIELQKTEATQLA | 40,0 | DENV-4-E | TLDFELTKTTAKEVA | 66,7 |
| DENV-1-E | DIELLKTEVTNPAVL | 20,0 | DENV-2-E | DFELIKTEAKQPATL | 33,3 | DENV-3-E | DIELQKTEATQLATL | 26,7 | DENV-4-E | DFELTKTTAKEVALL | 53,3 |
| DENV-1-E | ELLKTEVTNPAVLRK | 20,0 | DENV-2-E | ELIKTEAKQPATLRK | 20,0 | DENV-3-E | ELQKTEATQLATLRK | 26,7 | DENV-4-E | ELTKTTAKEVALLRT | 26,7 |
| DENV-1-E | LKTEVTNPAVLRKLC | 6,7 | DENV-2-E | IKTEAKQPATLRKYC | 20,0 | DENV-3-E | QKTEATQLATLRKLC | 0,0 | DENV-4-E | TKTTAKEVALLRTYC | 26,7 |
| DENV-1-E | TEVTNPAVLRKLCIE | 13,3 | DENV-2-E | TEAKQPATLRKYCIE | 20,0 | DENV-3-E | TEATQLATLRKLCIE | 6,7 | DENV-4-E | TTAKEVALLRTYCIE | 26,7 |
| DENV-1-E | VTNPAVLRKLCIEAK | 0,0 | DENV-2-E | AKQPATLRKYCIEAK | 0,0 | DENV-3-E | ATQLATLRKLCIEGK | 0,0 | DENV-4-E | AKEVALLRTYCIEAL | 13,3 |
| DENV-1-E | NPAVLRKLCIEAKIS | 6,7 | DENV-2-E | QPATLRKYCIEAKLT | 0,0 | DENV-3-E | QLATLRKLCIEGKIT | 0,0 | DENV-4-E | EVALLRTYCIEALIS | 60,0 |
| DENV-1-E | AVLRKLCIEAKISNT | 6,7 | DENV-2-E | ATLRKYCIEAKLTNT | 6,7 | DENV-3-E | ATLRKLCIEGKITNI | 13,3 | DENV-4-E | ALLRTYCIEALISNI | 33,3 |
| DENV-1-E | LRKLCIEAKISNTTT | 40,0 | DENV-2-E | LRKYCIEAKLTNTTT | 40,0 | DENV-3-E | LRKLCIEGKITNITT | 26,7 | DENV-4-E | LRTYCIEALISNITT | 13,3 |
| DENV-1-E | KLCIEAKISNTTTDS | 40,0 | DENV-2-E | KYCIEAKLTNTTTES | 46,7 | DENV-3-E | KLCIEGKITNITTDS | 40,0 | DENV-4-E | TYCIEALISNITTAT | 13,3 |
| DENV-1-E | CIEAKISNTTTDSRC | 33,3 | DENV-2-E | CIEAKLTNTTTESRC | 26,7 | DENV-3-E | CIEGKITNITTDSRC | 26,7 | DENV-4-E | CIEALISNITTATRC | 13,3 |
| DENV-1-E | EAKISNTTTDSRCPT | 53,3 | DENV-2-E | EAKLTNTTTESRCPT | 26,7 | DENV-3-E | EGKITNITTDSRCPT | 26,7 | DENV-4-E | EALISNITTATRCPT | 13,3 |
| DENV-1-E | KISNTTTDSRCPTQG | 20,0 | DENV-2-E | KLTNTTTESRCPTQG | 13,3 | DENV-3-E | KITNITTDSRCPTQG | 13,3 | DENV-4-E | LISNITTATRCPTQG | 0,0 |
| DENV-1-E | SNTTTDSRCPTQGEA | 6,7 | DENV-2-E | TNTTTESRCPTQGEP | 26,7 | DENV-3-E | TNITTDSRCPTQGEA | 20,0 | DENV-4-E | SNITTATRCPTQGEP | 20,0 |
| DENV-1-E | TTTDSRCPTQGEATL | 6,7 | DENV-2-E | TTTESRCPTQGEPSL | 13,3 | DENV-3-E | ITTDSRCPTQGEAVL | 26,7 | DENV-4-E | ITTATRCPTQGEPYL | 13,3 |

**Fig 2. Example showing several epitope hotspots (marked in red) found in the viral E protein of DENV-1, DENV-2, DENV-3, and DENV-4.**

of peptides with short (3 to 12) amino acid sequences that interact with antibodies based on their primary structure. Conformational epitopes, in turn, contain critical amino acids that are brought together by the folded protein to form more complex configurations recognized by antibodies. Here, we identified the linear epitope profile recognized by serum samples from dengue and Zika patients in the E and NS1 proteins of DENV and ZIKV. The serum samples demonstrated the ability to detect epitopes distributed throughout the entire viral protein, indicating no specific preference for any particular region. The detected epitopes exhibited a high degree of variability, an expected outcome given the high polymorphism of the Human Major Histocompatibility Complex (HMHC) genes responsible for presenting epitopes to the immune system [28].

Most of the detected peptides showed similar detection rates when analyzed with sera from dengue and Zika patients, *i.e.*, the antibodies derived from these patients presented high levels of cross-reactivity against the viral proteins. This observation aligns with previous findings

**Table 3. Identification of virus-specific peptides.**

| Virus | Protein | Peptide sequence | Sera detection rate (%) | | *p*-value |
|---|---|---|---|---|---|
| | | | Dengue | Zika | |
| DENV-2 | E | VHRQWFLDLPLPWLP | 22.0 | 0.0 | 0.0215 |
| DENV-2 | E | TQGEPSLNEEQDKRF | 32.5 | 4.76 | 0.0224 |
| DENV-4 | NS1 | TQTVGPWHLGKLEID | 35.0 | 9.52 | 0.0365 |
| ZIKV | E | LELDPPFGDSYIVIG | 5.0 | 28.57 | 0.0160 |

**Table 4. Virus-specific peptides within the epitope hotspots.**

| Virus | Protein | Sequence | Detection rate (%) |
|---|---|---|---|
| DENV-2 | E | TTTESRCPTQGEPSL | 13.3 |
| DENV-2 | E | TESRCPTQGEPSLNE | **53.3** |
| DENV-2 | E | SRCPTQGEPSLNEEQ | **40.0** |
| DENV-2 | E | CPTQGEPSLNEEQDK | **53.3** |
| DENV-2 | E | **TQGEPSLNEEQDKRF** | **73.3** |
| DENV-2 | E | GEPSLNEEQDKRFIC | **46.7** |
| DENV-2 | E | PSLNEEQDKRFICKH | 13.3 |
| DENV-4 | NS1 | QGYATQTVGPWHLGK | 0.0 |
| DENV-4 | NS1 | YATQTVGPWHLGKLE | 6.7 |
| DENV-4 | NS1 | **TQTVGPWHLGKLEID** | **53.3** |
| DENV-4 | NS1 | TVGPWHLGKLEIDFG | **53.3** |
| DENV-4 | NS1 | GPWHLGKLEIDFGEC | 13.3 |
| DENV-4 | NS1 | WHLGKLEIDFGECPG | 6.7 |
| ZIKV | E | STENSKMMLELDPPF | 20.0 |
| ZIKV | E | ENSKMMLELDPPFGD | **53.3** |
| ZIKV | E | SKMMLELDPPFGDSY | **53.3** |
| ZIKV | E | MMLELDPPFGDSYIV | **53.3** |
| ZIKV | E | **LELDPPFGDSYIVIG** | **40.0** |
| ZIKV | E | LDPPFGDSYIVIGVG | **33.3** |
| ZIKV | E | PPFGDSYIVIGVGEK | 26.7 |

The detection rate value of the peptides within the epitope hotspots is marked in bold letters.

reported in the literature [13–15], which also emphasized the existence of cross-reactivity among antibodies targeting these viral proteins. Among these peptides, we identified five with high detection rates: WEVEDYGFGVFTTNI, LELDFDLCEGTTVVV, DCEPRSGIDFNEMIL, PIVTDKEKPVNIETE, and TAGPWHLGKLEMDFD; therefore, they could be promising candidates for developing peptide-derived vaccines against both DENV and ZIKV. Falconi-Agapito and colleagues. described the peptide WEVEDYGFGVFTTNI as being part of an NS1 protein site containing B-cell epitopes for DENV-1, pan-DENV, and pan-flavivirus [15*], which supports our observations. However, we did not find studies describing the other peptides as B-cell epitopes in the available literature. Although the sera from dengue and Zika patients demonstrated high cross-immune reactivity against DENV and ZIKV peptides, we identified four virus-specific peptides (DENV-2 E protein: VHRQWFLDLPLPWLP and TQGEPSLNEEQDKRF; DENV-4 NS1 protein: TQTVGPWHLGKLEID; and ZKIKV E protein: LELDPPFGDSYIVIG). A previous study showed that peptides selected by bioinformatic analysis, including the entire or part of the DENV-2 E peptide VHRQWFLDLPLPWLP

**Table 5. Peptides detected by the serum samples from dengue and Zika patients at similar high detection rates.**

| Virus | Protein | Sequence | Sera detection rate (%) | | p-value |
|---|---|---|---|---|---|
| | | | Dengue | Zika | |
| DENV-3 | NS1 | WEVEDYGFGVFTTNI | 52.50 | 76.19 | 0.0997 |
| DENV-1 | NS1 | LELDFDLCEGTTVVV | 42.50 | 57.14 | 0.2963 |
| DENV-4 | E | DCEPRSGIDFNEMIL | 42.50 | 52.38 | 0.5901 |
| DENV-1 | E | PIVTDKEKPVNIETE | 40.00 | 52.38 | 0.4215 |
| DENV-2 | NS1 | TAGPWHLGKLEMDFD | 42.50 | 47.62 | 0.7889 |

**Table 6. Peptides with high detection rates within the epitope hotspots.**

| Virus | Protein | Sequence | Detection rate (%) |
|---|---|---|---|
| DENV-3 | NS1 | AWNVWEVEDYGFGVF | 6.7 |
| DENV-3 | NS1 | NVWEVEDYGFGVFTT | **60.0** |
| DENV-3 | NS1 | **WEVEDYGFGVFTTNI** | **40.0** |
| DENV-3 | NS1 | VEDYGFGVFTTNIWL | 26.7 |
| DENV-3 | NS1 | HLGKLELDFDLCEGT | 20.0 |
| DENV-3 | NS1 | GKLELDFDLCEGTTV | **40.0** |
| DENV-3 | NS1 | **LELDFDLCEGTTVVV** | 53.3 |
| DENV-3 | NS1 | LDFDLCEGTTVVVDE | 0.0 |

described in our study, were recognized by human sera [29]. This peptide can also serve as an epitope presented by HMHC I to T cells [30–34]. Another study demonstrated that individuals exposed to infections with each of the four DENV serotypes detected a bioinformatically predicted linear B-cell epitope (RCPTQGEPSLNEEQDKRF) of the DENV-2 E protein, which contains the peptide TQGEPSLNEEQDKRF described in our research [35]. This peptide can also serve as an epitope presented by HMHC I to T cells [30,36]. The peptide TQTVGPWHLGKLEID from the DENV-4 NS1 protein, found to be highly specific for DENV in the present study, was not described previously in the literature as a B-cell epitope. However, a previous study showed that a bioinformatically predicted pan-serotype sequence (GPWHLGKLE), which is a part of the specific sequence we found for DENV, activated T cells when using a peptide (AGPWHLGKLELDFNY) containing this pan-serotype sequence [37]. The ZIKV E protein epitope LELDPPFGDSYIVIG described in our study was not previously documented as a B-cell epitope in the literature. Nevertheless, this peptide can serve as an epitope presented by HMHC I to T cells [36,38]. These findings suggest that the four virus-specific peptides (DENV-2 E protein: VHRQWFLDLPLPWLP and TQGEPSLNEEQDKRF, DENV-4 NS1 protein: TQTVGPWHLGKLEID, and ZIKV E protein: LELDPPFGDSYIVIG) described in this study can be used for the development of diagnostic tests and vaccine candidates. However, they showed low detection rates in sub-arrays analyzed with individual serum samples. Based on these data and considering the variability of epitopes detected by pooled serum samples in our initial peptide microarray platform, it seems unlikely to achieve highly effective diagnostic tests or vaccines using a single peptide. These findings highlight the importance of considering a broader range of epitopes or using a multi-peptide approach to enhance the performance and reliability of such diagnostic tests or vaccine candidates. Nagar and colleagues evidenced the immunogenicity of several peptides selected by bioinformatics from the DENV E and NS1 proteins [29]. They suggested combining peptides in a multiple antigen format to improve the performance of diagnostic tests, a hypothesis in agreement with our findings.

During our analysis, we identified several epitopes located within epitope hotspots. These hotspots correspond to clusters of peptides detected in more than 30% of the sub-arrays during the initial screening. This observation emphasizes the crucial role of specific sites within the viral proteins in eliciting an immune response against these viruses. It also suggests the potential value of further investigating these regions in the context of vaccine development and diagnostic applications. We also analyzed the immunogenicity of several peptides from the epitope hotspots using individual serum samples in each sub-array. However, we could not include all peptides within the clusters for this analysis due to budget limitations. Our findings revealed the presence of four virus-specific peptides, three of which (DENV: TQGEPSLNEEQDKRF and TQTVGPWHLGKLEID; and ZIKV: LELDPPFGDSYIVIG) were located within the epitope hotspots. Additionally, two peptides (WEVEDYGFGVFTTNI and LELDFDLCEGTTVVV) within the epitope hotspots exhibited high detection rates by the serum samples of both dengue

**Table 7. BLAST comparison test for peptide similarity analysis against other DENV-1, DENV-2, DENV-3, DENV-4, and ZIKV strains.**

| Query | Virus—Protein | Query Cover (%)—Interval | Query Cover—Mean (%) | Query Cover—Median (%) | Per. Ident (%)—Interval | Per. Ident—Mean (%) | Per. Ident—Median (%) |
|---|---|---|---|---|---|---|---|
| VHRQWFLDLPLPWLP | DENV-1—E | 100.00–86.00 | 86.00 | 86.00 | 100.00–92.31 | 92.52 | 92.31 |
| | DENV-2—E | 100.00–100.00 | 100.00 | 100.00 | 100.00–100.00 | 100.00 | 100.00 |
| | DENV-3—E | 100.00–86.00 | 93.80 | 100.00 | 100.00–92.31 | 92.43 | 92.31 |
| | DENV-4—E | 86.00–86.00 | 86.01 | 86.00 | 100.00–92.31 | 92.33 | 92.31 |
| | ZIKV—E | 100.00–86.00 | 86.00 | 86.00 | 76.92–69.23 | 69.27 | 69.23 |
| TQGEPSLNEEQDKRF | DENV-1—E | 100.00–80.00 | 80.73 | 80.00 | 100.00–75.00 | 75.86 | 75.00 |
| | DENV-2—E | 100.00–100.00 | 100.00 | 100.00 | 100.00–100.00 | 100.00 | 100.00 |
| | DENV-3—E | 93.00–80.00 | 84.81 | 80.00 | 83.33–75.01 | 75.01 | 75.00 |
| | DENV-4—E | 100.00–80.00 | 94.02 | 100.00 | 83.33–73.33 | 83.32 | 83.33 |
| | ZIKV—E | 100.00–46.00 | 96.98 | 100.00 | 100.00–42.11 | 43.93 | 42.11 |
| TQTVGPWHLGKLEID | DENV-1—NS1 | 100.00–100.00 | 100.00 | 100.00 | 93.33–86.67 | 86.73 | 86.67 |
| | DENV-2—NS2 | 100.00–100.00 | 100.00 | 100.00 | 100.00–86.67 | 86.75 | 86.67 |
| | DENV-3—NS3 | 100.00–100.00 | 100.00 | 100.00 | 86.67–86.67 | 86.67 | 86.67 |
| | DENV-4—NS4 | 100.00–40.00 | 83.76 | 100.00 | 100.00–66.67 | 87.94 | 93.33 |
| | ZIKV—NS5 | 100.00–26.00 | 93.44 | 100.00 | 100–57.14 | 65.29 | 64.29 |
| LELDPPFGDSYIVIG | DENV-1—E | 100.00–93.00 | 99.63 | 100.00 | 92.86–78.57 | 78.88 | 78.57 |
| | DENV-2—E | 100.00–93.00 | 94.37 | 93.00 | 92.86–78.57 | 79.61 | 78.57 |
| | DENV-3—E | 100.00–93.00 | 99.91 | 100.00 | 78.57–71.43 | 71.45 | 71.43 |
| | DENV-4—E | 100.00–93.00 | 93.11 | 93.00 | 92.86–92.86 | 92.86 | 92.86 |
| | ZIKV—E | 100.00–100.00 | 100.00 | 100.00 | 100.00–100.00 | 100.00 | 100.00 |
| WEVEDYGFGVFTTNI | DENV-1—NS1 | 100.00–100.00 | 100.00 | 100.00 | 100.00–93.33% | 97.19 | 100.00 |
| | DENV-2—NS2 | 100.00–100.00 | 100.00 | 100.00 | 93.33–83.33 | 83.33 | 83.33 |
| | DENV-3—NS3 | 100.00–100.00 | 100.00 | 100.00 | 100.00–100.00 | 100.00 | 100.00 |
| | DENV-4—NS4 | 100.00–13.00% | 95.40 | 100.00 | 100.00–75.00 | 78.01 | 77.78 |
| | ZIKV—NS5 | 100.00–13.00% | 79.79 | 86.00 | 100.00–69.23 | 70.91 | 69.23 |
| LELDFDLCEGTTVVV | DENV-1—NS1 | 100.00–100.00 | 100.00 | 100.00 | 100.00–100.00 | 100.00 | 100.00 |
| | DENV-2—NS2 | 100.00–100.00 | 100.00 | 100.00 | 100.00–86.67 | 87.94 | 86.67 |
| | DENV-3—NS3 | 100.00–100.00 | 100.00 | 100.00 | 100.00–80.00 | 80.03 | 80.00 |
| | DENV-4—NS4 | 100.00–33.00 | 90.43 | 100.00 | 100.00–66.67 | 76.58 | 69.23 |
| | ZIKV—NS5 | 100.00–53.00 | 92.47 | 100.00 | 100.00–50.00 | 58.09 | 50.00 |
| DCEPRSGIDFNEMIL | DENV-1—E | 100.00–100.00 | 100.00 | 100.00 | 100.00–73.33 | 74.06 | 73.33 |
| | DENV-2—E | 100.00–100.00 | 100.00 | 100.00 | 100.00–66.67 | 66.80 | 66.67 |
| | DENV-3—E | 100.00–100.00 | 100.00 | 100.00 | 80.00–73.33 | 73.34 | 73.33 |
| | DENV-4—E | 100.00–100.00 | 100.00 | 100.00 | 100.00–100.00 | 100.00 | 100.00 |
| | ZIKV—E | 100.00–46.00 | 96.38 | 100.00 | 100.00–50.00 | 61.55 | 61.54 |
| PIVTDKEKPVNIETE | DENV-1—E | 100.00–100.00 | 100.00 | 100.00 | 100.00–100.00 | 100.00 | 100.00 |
| | DENV-2—E | 100.00–100.00 | 100.00 | 100.00 | 93.33–73.33 | 73.45 | 73.33 |

*(Continued)*

**Table 7.** (Continued)

| Query | Virus—Protein | Query Cover (%)—Interval | Query Cover—Mean (%) | Query Cover—Median (%) | Per. Ident (%)—Interval | Per. Ident—Mean (%) | Per. Ident—Median (%) |
|---|---|---|---|---|---|---|---|
| | DENV-3—E | 100.00–100.00 | 100.00 | 100.00 | 93.33–73.33 | 73.60 | 73.33 |
| | DENV-4—E | 60.00–13.00 | 23.63 | 26.00 | 100.00–57.14 | 98.70 | 100.00 |
| | ZIKV—E | 80.00–73.00 | 76.50 | 76.50 | 58.33–42.86 | 50.60 | 50.60 |
| TAGPWHLGKLEMDFD | DENV-1—NS1 | 100.00–100.00 | 100.00 | 100.00 | 100.00–93.33 | 93.41 | 93.33 |
| | DENV-2—NS2 | 100.00–100.00 | 100.00 | 100.00 | 100.00–100.00 | 100.00 | 100.00 |
| | DENV-3—NS3 | 100.00–100.00 | 100.00 | 100.00 | 93.33–86.70 | 86.68 | 86.67 |
| | DENV-4—NS4 | 100.00–40.00 | 98.63 | 100.00 | 100.00–50.00 | 87.55 | 85.71 |
| | ZIKV—NS5 | 93.00–26.00 | 48.33 | 26.00 | 100.00–66.67 | 88.89 | 100.00 |

and Zika patients. Epitope hotspots contain immunodominant epitopes recognized by a higher number of individuals when compared to 15-aa sequence peptides. Consequently, peptides with sequences containing all the epitope hotspots, spanning up to ~30 aa, have more potential than peptides of only 15 aa for developing diagnostic tests and vaccines. Mishra and colleagues found that a concatemer of an immunoreactive 20-aa ZIKV NS2B peptide (49 aa) demonstrated better performance than shorter peptides in immunoassays with Zika patients' sera [39], which supports our hypothesis that using the epitope hotspot sequences instead of 15-aa peptides can improve the performance of serological diagnostic tests and vaccines. Our findings regarding B-cell epitope hotspots in the DENV and ZIKV proteins are consistent with an earlier study conducted by Kam and colleagues [40], in which the authors identified numerous hotspots where antibodies bind within ZIKV antigens. Other viruses, such as HIV and SARS coronavirus, have also been described as presenting T-cell epitope hotspots [41–45]. These clusters or T-cell epitope hotspots are suitable for epitope-based vaccine development since they often encompass multiple promiscuous epitopes capable of binding to various alleles of HLA supertypes, thus maximizing population coverage [37]. The studies mentioned above collectively support the concept of epitope hotspots, highlighting specific regions within viral proteins that trigger robust immune responses from both B and T cells. In the future, we intend to explore the potential of using peptides containing the epitope hotspot sequences. By incorporating these specific epitopes, we aim to enhance the effectiveness and specificity of our diagnostic tests and vaccine development efforts.

A limitation of our study was the restricted number of individuals in the control group. In dengue-endemic regions, such as Brazil and Paraguay, locating individuals unexposed to DENV is challenging. Despite this limitation, epitope hotspot identification within viral proteins has provided valuable insights and instilled confidence in the results obtained in this study. Kam and colleagues [40] also mapped the E and NS1 proteins of ZIKV using sera from ZIKV-infected adult patients. They used 32 serum samples from anti-ZIKV IgG-positive patients to test against a peptide library derived from the yellow fever virus for background subtraction. The authors found low background reactivity to YFV, which did not interfere with ZIKV epitope identification. This literature data further supports the reliability of our findings, even in light of the low number of control serum samples used. Indeed, future studies conducted using control groups with a larger number and more diverse individuals would be beneficial to validate and reinforce the findings obtained herein.

In summary, our study has yielded valuable insights into the linear epitope profiles recognized by serum samples from patients with dengue and Zika infections, particularly within the

E and NS1 proteins of DENV and ZIKV. These findings contribute to our understanding of the immune response to these viruses and may have potential implications for diagnostic tests and vaccine development. We identified several epitopes within epitope hotspots, indicative of specific regions within the viral proteins that elicit strong immune responses. Among these epitope hotspots, we discovered two DENV-specific peptides (TQGEPSLNEEQDKRF and TQTVGPWHLGKLEID) and one ZIKV-specific peptide (LELDPPFGDSYIVIG). However, they exhibited low detection rates in our assays. On the other hand, we also identified two peptides (WEVEDYGFGVFTTNI and LELDFDLCEGTTVVV) within the epitope hotspots that showed high detection rates. Based on these findings, we propose that peptides encompassing all epitope hotspots, spanning approximately 30 amino acids, hold promising potential for use in diagnostic tests and vaccine development.

## Supporting information

**S1 Checklist. STROBE statement—Checklist of items that should be included in reports of observational studies.**
(DOCX)

**S1 Table. Epitope mapping with sera of DENV-infected individuals.**
(XLSX)

**S2 Table. Epitope mapping with sera of ZIKV-infected individuals.**
(XLSX)

**S3 Table. Identification of epitope hotspots.**
(XLSX)

**S4 Table. List of peptides included in the microarray platform for specificity analysis.**
(XLSX)

**S5 Table. Comparison of peptide detection rates with sera from dengue and Zika patients.**
(XLSX)

## Author Contributions

**Conceptualization:** Victor Hugo Aquino, Luiz Tadeu Moraes Figueiredo.

**Data curation:** Victor Hugo Aquino.

**Formal analysis:** Victor Hugo Aquino, Marcilio J. Fumagalli, Angélica Silva, Cynthia Bernal, Luiz Tadeu Moraes Figueiredo.

**Funding acquisition:** Victor Hugo Aquino, Yvalena Guillen, Luiz Tadeu Moraes Figueiredo.

**Investigation:** Marcilio J. Fumagalli, Angélica Silva, Bento Vidal de Moura Negrini, Alejandra Rojas, Cynthia Bernal.

**Methodology:** Marcilio J. Fumagalli, Angélica Silva, Bento Vidal de Moura Negrini, Alejandra Rojas, Cynthia Bernal.

**Project administration:** Victor Hugo Aquino.

**Resources:** Bento Vidal de Moura Negrini, Yvalena Guillen, Luiz Tadeu Moraes Figueiredo.

**Supervision:** Victor Hugo Aquino, Yvalena Guillen.

**Writing – original draft:** Victor Hugo Aquino.

**Writing – review & editing:** Victor Hugo Aquino, Marcilio J. Fumagalli, Angélica Silva, Bento Vidal de Moura Negrini, Alejandra Rojas, Yvalena Guillen, Cynthia Bernal, Luiz Tadeu Moraes Figueiredo.

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
