## [Decision Letter · Decision Letter 0]

17 Aug 2023

PONE-D-23-18775Linear epitope mapping in the E and NS1 proteins of dengue and Zika viruses: prospection of peptides for vaccines and diagnostics.PLOS ONE

Dear Dr. Aquino,

Thank you for submitting your manuscript to PLOS ONE. After careful consideration, we feel that it has merit but does not fully meet PLOS ONE’s publication criteria as it currently stands. Therefore, we invite you to submit a revised version of the manuscript that addresses the points raised during the review process.

In detail, two experts in the field have accurately evaluated your manuscript and both of themn suggest you to undertake a revision in order to make your work acceptable for publication. In detail, I agree with the suggestion that the overall data presentation could be ameliorated and that quite a few negative control samples have been used. So, in the unlucky case you could not undertakle the suggested additional experiments, at least go deep into the discussion section trying to convice the reader (and myself...) that the data are in anyway a good support for your conclusions.

We look forward to receiving your revised manuscript.

Kind regards,

Vittorio Sambri, M.D., Ph.D.

Academic Editor

PLOS ONE

3. Please upload a new copy of Figure 2 as the detail is not clear. Please follow the link for more information: https://blogs.plos.org/plos/2019/06/looking-good-tips-for-creating-your-plos-figures-graphics/" https://blogs.plos.org/plos/2019/06/looking-good-tips-for-creating-your-plos-figures-graphics/

4. We suggest you thoroughly copyedit your manuscript for language usage, spelling, and grammar. If you do not know anyone who can help you do this, you may wish to consider employing a professional scientific editing service.

Reviewers' comments:

Reviewer's Responses to Questions

**Comments to the Author**

1. Is the manuscript technically sound, and do the data support the conclusions?

Reviewer #1: Yes

Reviewer #2: Partly

2. Has the statistical analysis been performed appropriately and rigorously? 

Reviewer #1: Yes

Reviewer #2: Yes

3. Have the authors made all data underlying the findings in their manuscript fully available?

Reviewer #1: Yes

Reviewer #2: Yes

4. Is the manuscript presented in an intelligible fashion and written in standard English?

Reviewer #1: Yes

Reviewer #2: Yes

5. Review Comments to the Author

Reviewer #1: In the manuscript by Aquino et al., the authors rigorously map dengue and Zika virus protein epitopes that are found in serum of infected patients. The identify epitope hot spots and delineate peptides that are found in these regions that may be used for improved diagnostic tests and vaccine development. The manuscript is well-written and the results are explained well. My minor comments focus on presentation of the data, which needs to be improved for clarity to the reader.

It would be helpful if the authors better described to the reader how to interpret Fig 1 and Table 2 together. Since the table and figure contain similar data, what is the purpose of including each? With a better explanation, it would make more sense to use both. Also, in Figure 1, it should be more clearly stated in the caption that the vertical bars represent patient samples. In Table 2, is the formatting is incorrect, or do some patients in the Zika population not have P and E data? There are blank rows.

For Table 7, it would be helpful if the authors highlighted rows with significant results or peptides for the reader to pay attention to.

Table 4 and 6 – What is the purpose of including a column called “code?” This is not defined in the text.

Figure 2 is nearly illegible. Can the authors reduce the redundancy of text by combing cells that contain the same text? Also, what is the purpose of the “code” column? The figure needs to be simplified or split into multiple panels.

Table 6 – why are the p-values such high numbers, greater than 1?

Line 68 – “practically the entire country” is vague and colloquial. This should be more precise.

Reviewer #2: The manuscript PONE-D-23-18775 "Linear epitope mapping in the E and NS1 proteins of dengue and Zika viruses: prospection of peptides for vaccines and diagnostics" describes the mapping of four novel peptides which are suitable candidates for developing peptide-derived vaccines and diagnostic tests specific to DENV and ZIKV. They propose that larger sequences, instead of the peptides alone, containing the entire epitope hot spots would enhance the performance and reliability of such vaccines and diagnostic tests. My major concern is the low number of negative control serum samples used in this work. I would like to recommend a couple of additional experiments (comments 1&2), and a few minor corrections/inquiries (comments 3-9), which I believe will substantially improve the quality of the article.

1) Only 5 serum samples from YFV vaccinated and DENV/ZIKV seronegative healthy women were used as controls. From these, 4 or 3 serum samples were used for the epitope mapping or peptide microarray assays, respectively. To arrive to the conclusions the authors state in their work, a bigger and more representative (age-gender) population should be included as control group.

2) It is not clear to me how many DENV serum samples, from each serotype, have been used in the work. Authors should characterize and inform the number of samples collected and analyzed for each DENV serotype, in order to demonstrate similar sample sizes for each of the four DENV serotypes.

3) How do you establish the 30% threshold detection rate value for the determination of hot spots?

4) Line 36-37: Rephrase "Dengue and Zika patients detected..."

5) Line 148: "Dyligt" misspelt

6) Line 215-216: Correct "The average age was 33 and 34.4 years for dengue Zika..."

7) Table 6: "p-value" column shoud be "Detection rate (%)" instead

8) Line 348: "ZKIKV" misspelt

9) The Discussion section is too long. It should be shortened.

6. PLOS authors have the option to publish the peer review history of their article (what does this mean?). If published, this will include your full peer review and any attached files.

Reviewer #1: No

Reviewer #2: No

---

## [Author Response · Author response to Decision Letter 0]

19 Sep 2023

Answer to Editor and Reviewers 

 Editor 

 Answer 

The manuscript style was reviewed to meet PLOS ONE's style requirements. 

 Answer 

We have included the correct grant information: This study was financially supported by Fundação de Amparo à Pesquisa do Estado de São Paulo, FAPESP, Brazil (grant numbers 2017/09194-3 and 2019/26119-0), and Consejo Nacional de Ciencia y Tecnología, CONACYT, Paraguay (grant number PIRT19-1). The funders had no role in study design, data collection and analysis, decision to publish, or preparation of the manuscript. 

3. Please upload a new copy of Figure 2 as the detail is not clear. Please follow the link for more information: 

 Answer 

We uploaded an improved version of Figures 1 and 2. 

4. We suggest you thoroughly copyedit your manuscript for language usage, spelling, and grammar. If you do not know anyone who can help you do this, you may wish to consider employing a professional scientific editing service. 

 Answer 

The manuscript underwent professional copyediting by a scientific editing service, enhancing the overall linguistic quality. 

Reviewer #1: 

1- It would be helpful if the authors better described to the reader how to interpret Fig 1 and Table 2 together. Since the table and figure contain similar data, what is the purpose of including each? With a better explanation, it would make more sense to use both. Also, in Figure 1, it should be more clearly stated in the caption that the vertical bars represent patient samples. In Table 2, is the formatting is incorrect, or do some patients in the Zika population not have P and E data? There are blank rows. 

 Answer 

We have enhanced the clarity of our manuscript by revising the text that elaborates on the findings presented in Fig 1 and Table 2 (lines 235-240 and 254-258). Additionally, we have provided in the caption a comprehensive explanation to enhance the interpretation of Fig 1 (lines 242-245). 

The occurrence of empty rows within Table 2 resulted from initial formatting issues, which have been resolved. 

2- For Table 7, it would be helpful if the authors highlighted rows with significant results or peptides for the reader to pay attention to. 

 Answer 

We have highlighted the Percent Identity - Median (%) column values within Table 7, as they encompass the most pertinent results. 

3- Table 4 and 6 – What is the purpose of including a column called “code?” This is not defined in the text. 

 Answer 

Each peptide has a specific code within the microarray slide. However, we have removed the Code containing column from Tables 4 and 6 to avoid confusion. 

4- Figure 2 is nearly illegible. Can the authors reduce the redundancy of text by combing cells that contain the same text? Also, what is the purpose of the “code” column? The figure needs to be simplified or split into multiple panels. 

 Answer 

We uploaded an improved version of Figures 1 and 2. 

5- Table 6 – why are the p-values such high numbers, greater than 1? 

 Answer 

Apologies for the error. That specific column corresponds to the Detection rate (%) value, which has now been rectified. 

6- Line 68 – “practically the entire country” is vague and colloquial. This should be more precise. 

 Answer 

The manuscript underwent professional copyediting by a scientific editing service, enhancing the overall linguistic quality. 

Reviewer #2: 

1) Only 5 serum samples from YFV vaccinated and DENV/ZIKV seronegative healthy women were used as controls. From these, 4 or 3 serum samples were used for the epitope mapping or peptide microarray assays, respectively. To arrive to the conclusions the authors state in their work, a bigger and more representative (age-gender) population should be included as control group. 

 Answer 

We agree with the reviewer's suggestion regarding the potential benefits of increasing the number of control samples to enhance the reliability of our study. As acknowledged in our manuscript, the limitation of our study lies in the constrained availability of resources, preventing us from conducting additional microarray assays. Nonetheless, we maintain confidence in the presented results. We have revised the discussion section to incorporate relevant findings from the literature that further support our conclusions (lines 428-441). 

2) It is not clear to me how many DENV serum samples, from each serotype, have been used in the work. Authors should characterize and inform the number of samples collected and analyzed for each DENV serotype, in order to demonstrate similar sample sizes for each of the four DENV serotypes. 

 Answer 

The information regarding the DENV serotype at the time of recruitment has been incorporated into Table 1. However, it's important to note that we did not include an analysis of this variable in our study. This decision stems from the fact that in countries with DENV transmission, such as Brazil and Paraguay, individuals often encounter multiple DENV serotype infections throughout their lives. The identification of these infections can be a formidable challenge. The widespread prevalence of multi-serotype infections introduces a significant layer of complexity when attempting to explore potential associations between DENV epitope detection and the specific DENV serotype individuals had at the time of recruitment. 

3) How do you establish the 30% threshold detection rate value for the determination of hot spots? 

 Answer 

We rephrase the text for clarification (lines 262-264). 

"When analyzing peptide detection rates, we identified epitope hotspots constituted by clusters of peptides that consistently exceeded a 30% detection rate". 

4) Line 36-37: Rephrase "Dengue and Zika patients detected..." 

 Answer 

The manuscript underwent professional copyediting by a scientific editing service, enhancing the overall linguistic quality. 

5) Line 148: "Dyligt" misspelt 

 Answer 

The spelling of "Dyligt" was corrected. 

6) Line 215-216: Correct "The average age was 33 and 34.4 years for dengue Zika..." 

 Answer 

The statement was corrected (lines 219-220). 

7) Table 6: "p-value" column shoud be "Detection rate (%)" instead 

 Answer 

We apologize for the error that occurred. We have rectified the issue. 

8) Line 348: "ZKIKV" misspelt 

 Answer 

The spelling of "ZKIKV" was corrected. 

9) The Discussion section is too long. It should be shortened. 

Since the PLOS ONE journal does not impose restrictions on manuscript length, we respectfully seek the reviewer's approval to maintain the expanded Discussion section. We have revised this section to improve its overall clarity. Furthermore, we have included an additional statement to strengthen our findings and acknowledge the study's limitation arising from the limited number of control samples as mentioned above.

---

## [Editor Report · Decision Letter 1]

20 Sep 2023

Linear epitope mapping in the E and NS1 proteins of dengue and Zika viruses: prospection of peptides for vaccines and diagnostics.

PONE-D-23-18775R1

Dear Dr. Aquino,

We’re pleased to inform you that your manuscript has been judged scientifically suitable for publication and will be formally accepted for publication once it meets all outstanding technical requirements.

Kind regards,

Vittorio Sambri, M.D., Ph.D.

Academic Editor

PLOS ONE
---

## [Editor Report · Acceptance letter]

25 Sep 2023

PONE-D-23-18775R1 

Linear epitope mapping in the E and NS1 proteins of dengue and Zika viruses: prospection of peptides for vaccines and diagnostics. 

Dear Dr. Aquino:

I'm pleased to inform you that your manuscript has been deemed suitable for publication in PLOS ONE. Congratulations! Your manuscript is now with our production department. 

Kind regards, 

on behalf of

Professor Vittorio Sambri 

Academic Editor

PLOS ONE